# Natural Killer Cells in Chronic Lymphocytic Leukemia: Functional Impairment and Therapeutic Potential

**DOI:** 10.3390/cancers14235787

**Published:** 2022-11-24

**Authors:** Max Yano, John C. Byrd, Natarajan Muthusamy

**Affiliations:** 1Medical Science Training Program, The Ohio State University College of Medicine, Columbus, OH 43210, USA; 2Department of Internal Medicine, University of Cincinnati, Cincinnati, OH 45267, USA; 3Division of Hematology, Department of Internal Medicine, The Ohio State University, Columbus, OH 43210, USA

**Keywords:** cancer, leukemia, CLL, chronic lymphocytic leukemia, NK cell, natural killer cell, immunotherapy, adoptive cellular therapy, cell therapy

## Abstract

**Simple Summary:**

Natural killer (NK) cells are immune cells with potent anti-tumor and anti-infection activity. The potential benefit of NK cell therapy against chronic lymphocytic leukemia (CLL) has long been recognized, but efforts to develop effective NK cell therapies have been hampered by multiple factors including the immunosuppressive effects of CLL against NK cells. In this review, we first outline the specific NK cell impairment seen in CLL and the known mechanisms causing these defects. We then discuss the NK-altering effects of current CLL therapeutics as well as the past and present progress towards developing NK cell therapy for CLL.

**Abstract:**

Immunotherapy approaches have advanced rapidly in recent years. While the greatest therapeutic advances so far have been achieved with T cell therapies such as immune checkpoint blockade and CAR-T, recent advances in NK cell therapy have highlighted the therapeutic potential of these cells. Chronic lymphocytic leukemia (CLL), the most prevalent form of leukemia in Western countries, is a very immunosuppressive disease but still shows significant potential as a target of immunotherapy, including NK-based therapies. In addition to their antileukemia potential, NK cells are important immune effectors in the response to infections, which represent a major clinical concern for CLL patients. Here, we review the interactions between NK cells and CLL, describing functional changes and mechanisms of CLL-induced NK suppression, interactions with current therapeutic options, and the potential for therapeutic benefit using NK cell therapies.

## 1. Background

### 1.1. Introduction to CLL and Its Immunosuppressive Effects

Chronic lymphocytic leukemia (CLL) is the most prevalent adult leukemia, estimated to affect over 20,000 newly diagnosed patients and cause over 4000 deaths in the US in 2021 and representing 22–30% of all leukemia cases [1,2]. CLL is characterized by an accumulation of malignant small mature B lymphocytes, and while it is most commonly diagnosed in asymptomatic adults with lymphocytosis, it eventually progresses to symptoms that include fatigue, weight loss, lymphadenopathy, infections, bleeding, and anemia [3].

One of the key phenotypes produced by CLL is a significant immune deficiency. This leads to a high risk of infections that begins in patients with monoclonal B-cell lymphocytosis (a precursor to CLL), increases with progression to early leukemia, and is often exacerbated by CLL-directed therapies [4,5,6]. This risk includes a variety of pathogens at multiple sites, and the spectrum of risk varies by disease progression and treatment [7]. In addition to infection, CLL causes an increased risk of secondary malignancies, and CLL is associated with decreased survival from these cancers, which are a common cause of non-disease-related death in CLL patients [8,9,10,11]. This increased cancer risk is largely attributed to ineffective immune surveillance. CLL-induced immune dysregulation also manifests as a high risk of autoimmunity, predominantly autoimmune cytopenias [12,13].

CLL-induced immune dysfunction has been described in various components of the immune system, including dendritic cells [14], monocytes [15,16,17], neutrophils [15,18,19,20], systemic cytokines [21], complement [22,23,24,25], myeloid-derived suppressor cells [26,27], T cells [28,29], B cells [30,31], plasma cells [32], innate lymphoid cells [33], and immunoglobulin levels [34,35,36,37]. Many of these immune components are co-opted by CLL to not only avoid immune surveillance but to actually support the growth and survival of leukemic cells [38,39,40,41,42]. These immunosuppressive effects have been ascribed to multiple mechanisms, including inhibitory ligands, cytokines, metabolic changes, and adenosine signaling [43,44]. Interestingly, at least one early study suggested that NK cells themselves may contribute to this immune dysfunction by suppressing B cell immunoglobulin production [45]. Multiple studies have demonstrated a significant NK dysfunction (especially decreased cytotoxic activity) to accompany the other immune changes in CLL.

### 1.2. Natural Killer Cells

Natural killer (NK) cells are a type of lymphoid cell with innate immune function and represent 2–18% of total lymphocytes in the blood [46]. NK cells were first identified by their ability to lyse tumor cells, and learning to harness this capability has been an active area of research since the 1970s. The extensive preclinical research establishing the anti-tumor effects of NK cells has been additionally supported by clinical research demonstrating that people with low baseline NK activity in blood have an increased long-term risk of cancer and that NK infiltration into various tumors predicts a better prognosis [47,48,49,50,51]. In addition to their cytotoxic functions, NK cells release cytokines (notably, IFNγ and TNFα) and other factors that influence other adaptive and innate immune cells [52].

Natural killer cells are generally divided into two main subsets based on their expression of CD56 and CD16. CD56^bright^CD16^−^ NK cells (stage 4 NKs according to the developmental classification) are the major subset in secondary lymphoid tissues and have a primarily immunoregulatory function through cytokine secretion but have low cytotoxic activity [46,53]. In contrast, CD56^dim^CD16^+^ NK cells (stage 5 NKs) represent about 90% of NK cells in the blood and have potent cytotoxic activity with lesser cytokine secretion [46,53]. Even within these two major subsets, there is a wide variety of NK cell phenotypes, based on their combinatorial expression of various receptors for target recognition—one study using mass cytometry to measure NK cell diversity estimated 6000 to 30,000 individual NK cell phenotypes per person [54]. Among these, there is an established developmental hierarchy in which less mature cells express the receptor NKG2A and more mature cells express KIRs (killer-cell immunoglobulin-like receptors) and CD57 [55]. These immunophenotypic changes reflect functional differences in NK cells’ ability to proliferate, secrete cytokines, and lyse target cells [55,56]. While NK cells are broadly considered part of the innate immune system, certain stimuli are known to induce a population of persistent, memory-like NK cells with an enhanced response to repeated stimulation [57].

Natural killer cells recognize their target cells through a variety of innate germline-encoded receptors with activating or inhibitory effects. Notable among these are the KIRs, which sense specific HLA (human leukocyte antigen) class I alleles. KIRs with long cytoplasmic tails exert inhibitory effects via ITIM signaling, while KIRs with short cytoplasmic tails associate with DAP12 to transmit an activating signal via its ITAM [58]. KIRs are polymorphic, and there is significant heterogeneity among individuals’ KIR expression, which may contribute to their ability to recognize malignant cells in the context of cell therapy [58]. KIRs are also involved in the process of NK licensing, by which only the NKs capable of binding self-HLA gain full effector function [59]. Other activating receptors expressed by NK cells include CD16 (which recognizes IgG), NKG2D (binding MICA, MICB, ULBP1-6), NKG2C (HLA-E), NKp30 (BAT3, B7-H6), NKp44 (various), NKp46 (various), and DNAM-1 (CD112, CD155) [46,59,60,61,62]. Key inhibitory receptors include NKG2A (HLA-E), ILT-2 (MHC-I, HLA-G), and TIGIT (CD112, CD155) [46,59,60,61]. NK activation in response to a potential target is determined by the balance of activating/inhibiting signals from these receptors. Many of the ligands detected by these receptors are modulated by various forms of cellular stress, enabling NK cells to activate in response to malignancy, viral infection, or other dysfunctional states in the target cell [61]. Notably, CD16 stimulation alone is sufficient to activate NK cells, whereas other receptors must be activated together to achieve NK activation [59].

After target cell recognition, NK cells are capable of lysing the cell through two distinct mechanisms. The first is through the release of lytic granules, which notably contain perforin and granzyme B (along with other granzymes, serglycin, granulysin, and other proteins). Perforin is a pore-forming toxin that enables granzyme B to enter the target cell; granzyme B then activates apoptotic signaling in the target cell by cleaving caspase 3 and Bid, along with a wide array of intracellular proteins [63]. The second cytotoxic mechanism is through the expression of TRAIL, FasL, or TNFα. These pro-death signaling ligands bind to their corresponding receptor on the target cell, triggering the formation of the death-inducing signaling complex (DISC), which activates apoptotic signaling by cleaving caspases 8 and 10 [63].

### 1.3. Evidence for NK Cell Importance in Leukemia

Some of the earliest evidence for the role of NK cells against leukemia came in the context of hematopoietic stem cell transplant (HSCT). Ruggeri et al. demonstrated a lower risk of relapse after HSCT in cases with a graft-versus-host KIR-ligand mismatch (i.e., the donor NK cells expressed a KIR that was not inhibited by the recipient’s HLAs) [64]. Since this time, multiple models of beneficial KIR-HLA mismatch have been developed. The “missing ligand”/“receptor–ligand mismatch” model looks for donor KIRs that are not inhibited by the recipient HLA alleles, the “ligand–ligand mismatch” model looks for differences between HLA alleles expressed on the donor and recipient cells, and the “KIR-B” model looks for the presence of the KIR B haplotype on donor cells, which includes the expression of activating KIR receptors. Numerous subsequent studies have provided mixed evidence for the presence or absence of benefits from various models of KIR activation [65]. Such extensive contradictory evidence reflects the complexity of HSCT—any overarching conclusion from this large swath of evidence is complicated by differences in types of transplants, pre/posttransplant treatments, diseases, patient populations, and models of KIR/HLA mismatch studied.

For CLL, allogeneic HSCT can be an effective treatment with curative potential; autologous transplant improves progression-free survival but not overall survival and is generally not thought to be a curative therapy [66]. However, with the advanced age of most CLL patients, the high-frequency and long-term morbidity of graft-versus-host disease, and the effectiveness of less intensive treatments, allogeneic transplants are rarely pursued. A study of 537 allogeneic transplant patients demonstrated no influence from KIR genotypes on HSCT outcomes in CLL [67]. On the other hand, CLL patients are more likely than healthy donors to have genotypes containing a higher ratio of inhibitory to activating KIRs, which may imply that NK surveillance has a role in CLL development [68].

Monoclonal antibody therapies targeting CD20 (rituximab, ofatumumab, and obinutuzumab) and CD52 (alemtuzumab) have a proven track record of efficacy for CLL treatment. Leukemia-targeting antibodies have several mechanisms of action, including complement-dependent cytotoxicity, direct cytotoxicity, antibody-dependent cellular phagocytosis (ADCP), and antibody-dependent cellular cytotoxicity (ADCC) by NK cells [69,70]. The contribution of NK cells has been explored in clinical data by studying a polymorphism of CD16 (FcγRIIIA-158V) that binds IgG with higher affinity and improves NK activation versus the FcγRIIIA-158F allele [71,72]. While the presence of the 158V allele has been shown to improve responses to rituximab in follicular lymphoma and Waldenstrom’s macroglobulinemia [73,74,75], data in CLL have shown that CD16 polymorphisms do not predict a response to rituximab monotherapy or FCR (combination fludarabine, cyclophosphamide, and rituximab) [76,77].

Multiple types of NK cell therapy have been evaluated in other hematologic malignancies. For example, Miller et al. treated AML patients with haploidentical NK cells (stimulated overnight with IL-2) after fludarabine + cyclophosphamide conditioning, finding that NK cell therapy was safe and produced complete responses in several patients [78]. Choi et al. stimulated haploidentical donor-derived NKs with IL-15 and IL-21 and administered them after HSCT [79]. They found no acute toxicity from NK cells, plus a decreased risk of leukemia progression compared with a historical control [79]. Romee et al. stimulated haploidentical NKs with IL-12, IL-15, and IL-18 and administered them to patients with AML after fludarabine + cyclophosphamide conditioning, again finding several complete responses [80]. While none of these studies were designed or powered to determine any clinical benefit from the various treatments used, the preliminary data do provide intriguing hints of the efficacy of natural killer cells as a cellular therapy.

## 2. NK Dysfunction in CLL

### 2.1. Counts and Cytotoxicity

Studies consistently demonstrate increased counts of peripheral blood natural killer cells in CLL [81,82,83,84,85,86,87,88,89,90,91,92,93]. Their prognostic significance has been evaluated twice with a higher ratio of NK cells to CLL cells (but not absolute count), predicting a longer time to treatment in one report and absolute NK count correlating to overall survival in another [88,94]. Although one study showed the expansion of the CD56^dim^CD16^+^ NK subset [92], and one showed a relative decrease in this subset [95], most evidence suggests that CD56-bright/dim ratios do not change in CLL [83,91,96,97,98]. No consistent correlation has been found between NK counts and the Rai/Binet stage or the known prognostic factors in CLL (CD38 expression, Zap70 expression, IGHV mutation, or FISH results) [88,91,92,93,94].

In addition to increased counts, decreased cytotoxicity has been a consistent finding among most studies of NK cells in CLL [83,93,95,98,99,100,101,102,103,104,105,106,107,108,109], although a few studies have found normal levels of direct cytotoxicity or ADCC caused by patient-derived NKs [91,96,97,98]. This decreased function is accompanied by decreased NK granularity and the decreased expression of cytotoxicity-related genes, and it is thought to be independent of any disease stage [95,107,109].

### 2.2. NK Phenotypes

Experiments studying the expression of specific activating and inhibiting receptors on NK cells in CLL patients have led to varying and often conflicting results (Figure 1). Overall, Le Garff-Tavernier et al. [91] argue that NK phenotypes in CLL (specifically studying the CD16^+^CD56^dim^ subset) are broadly normal; most other studies do find alterations in NK receptor expression, although the specific differences are not consistent between studies. For example, five studies have found decreased NKG2D expression [83,93,95,98,110], while three have found normal expression [91,97,111]. Similarly conflicting results have been found for other activating NK receptors: NKp30 [84,91,92,95,97,98], NKp46 [91,92,95,97], NKG2C [91,98,112], and DNAM-1 [95,97]. Other activating receptors such as 2B4, NKp44, and NKp80 have been studied and found to be unchanged between patient and donor NKs [91,92,97]. Overall, multiple studies suggest that the downregulation of NK activating receptors may contribute to NK dysfunction in CLL patients; however, this finding has not been consistently replicated.

Regarding inhibitory NK receptors, evidence is similarly conflicting but may point to the upregulation of ILT-2 [91,113] and the downregulation of KIRs [83,91,97,98]. Additionally, immune checkpoints TIM3, BTLA, GITR, and LAG3 have been found to be upregulated in CLL patient NK cells [84,114,115,116]. NKG2A, LAIR-1, and KLRG1 have been studied and found unchanged between patient and donor NKs [91,98,117]. Similar to activating receptors, multiple studies implicate the upregulation of inhibitory receptors, but this has not been consistent, and more research will be beneficial to better characterize these changes.

The activation status of NKs in CLL patients is also controversial—Eskelund et al. used increased CD25 expression to suggest a chronic activation state, while Le Garff-Tavernier et al. interpreted decreased CD69 expression to suggest anergy [91,96]. MacFarlane et al. used an increase in CD27^+^ and a decrease in KIR^+^ NKs to argue for a loss of mature NK cells, while Hofland et al. used increased CD57 to argue for increased CD56^dim^ NK maturity [83,98]. A potential antileukemic NK population in CLL has been identified with a CD45RA^+^RO^+^ phenotype as well as CD19^+^ (suggesting trogocytosis); this population is increased in CLL patients and may suggest an active antileukemia NK response [118,119].

Finally, two studies have compared CLL samples to small lymphocytic lymphoma (SLL, a lymph node-predominant phenotype of early-stage CLL). MacFarlane et al. found that, other than a lack of NK expansion, the changes seen in SLL samples were similar to those seen in CLL samples (with some changes having greater or lesser magnitude than CLL; most did not reach statistical significance when compared to healthy donors) [83]. In contrast, Parry found no phenotypic differences between normal donor and SLL NKs [95]. In their microarray data, there were significant differences between normal donor, SLL, and CLL samples, but there was little overlap between the SLL and CLL signatures, suggesting major differences in the NK response to these disease phenotypes [95]. These studies reached significantly different conclusions, and further study may be necessary to clarify the effects of SLL on NK cell phenotypes.

## 3. Mechanisms of NK Dysfunction in CLL

Many mechanisms have been proposed to explain the NK cell dysfunction seen in CLL patients (Figure 2). These include the altered expression of NK cell activating/inhibiting ligands, the shedding of NK cell ligands, cytokine secretion, and changes in the immune microenvironment. Hofland et al. demonstrated that a 48 h coculture with CLL cells was sufficient to suppress NK cytotoxic function, suggesting a direct inhibitory effect of CLL on NK cells [98]. Additionally, Burton et al. and Reiners et al. demonstrated that supernatant from CLL cells or plasma from CLL patients could inhibit NK activity, implicating soluble factors produced by CLL cells in NK dysfunction [120,121]. In addition to its direct effects on NK cells, CLL induces numerous changes in the immune system, which are likely to have additional effects on NK cells [43].

### 3.1. Receptor–Ligand Interactions

As discussed above, natural killer cells identify targets through the expression of various ligands on the target cell surface. Different patterns of ligand expression, therefore, can influence a cell’s susceptibility to NK-mediated killing. One of the primary activating receptors for NK target identification is NKG2D, which recognizes several ligands in cancer cells. When Hilpert et al. examined CLL cells for NKG2D ligand expression (MICA, MICB, ULBP1, ULBP2, ULBP3), they found that 85% of CLL patients expressed at least one ligand, and most expressed more than one; however, when Veuillen et al. compared these to normal B cells, they found only ULBP3 to be upregulated [97,110]. Nückel et al. only looked at MICA and found it to be upregulated compared with normal B cells [111]. Expression levels of other NKG2D ligands are currently unknown in CLL. Veuillen et al. further looked at the expression of the DNAM-1 ligands CD112 and CD155, the CD244 ligand CD48, and the NKp30 ligand B7-H6 and found that these ligands are all expressed at similar levels on CLL cells as normal B cells [97]. The expressions of other NK-activating ligands for receptors such as NKp44, NKp46, and NKp80 are not yet known. Most patients’ CLL cells downregulate the receptors SLAMF1 and SLAMF7 (which bind homotypically to the same receptors on NK cells); these patients’ NK cells have impaired degranulation relative to those with retained SLAMF receptor expression, suggesting that SLAMF receptor downregulation impairs NK function [122,123]. Overall, these data suggest low levels of NK activating ligands in CLL cells, potentially leading to a lack of NK recognition (Figure 1).

Conversely, NK reactivity can be inhibited by the expression of select inhibitory ligands on target cells. Foremost among these are the MHC-I (major histocompatibility complex class I; HLA-A, HLA-B, and HLA-C) molecules that bind to KIRs and mediate the “missing self” model of regulating NK activation. One early paper compared CD5^+^ leukemic cells with CD5^−^ normal B cells in the same patient and found that leukemia cells had increased levels of MHC-I, which would inhibit NK cell cytotoxicity [124]. However, several more recent papers have compared CLL cells to normal B cells from healthy volunteers and found a decreased (although not absent) expression of MHC-I in CLL cells [97,125,126]. CLL cells overexpress HLA-E, which binds to the inhibitory receptor NKG2A or activating receptor NKG2C on NKs [97,117]. Accordingly, the blockade of NKG2A with monalizumab increases the NK cytotoxicity of CLL cells [117]. Another established ligand is HLA-G, which binds to the inhibitory receptors ILT-2 and KIR2DL4 on NKs. Results from this molecule have been discordant, with some studies finding no expression on CLL cells [127,128,129], while others did find HLA-G on CLL cells, at levels similar to or greater than normal B cells [97,130,131,132,133,134,135]. Interestingly, HLA-G has also been studied as a prognostic marker for CLL—two small studies found a correlation with lower HLA-G, leading to longer progression-free survival or time to treatment, but a larger study found no correlation with progression-free survival [130,133,134]. The blockade of HLA-G or ILT-2 in vitro increases the NK cytotoxicity of CLL cells [113,131]. GITRL is also expressed by CLL, and GITR blockade increases the NK activation and cytotoxicity of CLL cells in vitro [115]. Siglec-7 is an inhibitory glycan receptor expressed in NK cells; its ligands have been detected at increased levels in CLL cells [136]. Finally, CLL cells express ligands PD-L1, CD200, B7-H3, and HVEM, which are best known for their T cell-suppressive effects but also inhibit NK function [114,137,138,139,140]. Collectively, these studies demonstrate the broad upregulation of inhibitory ligands on CLL cells, contributing to CLL-induced NK dysfunction and the evasion of NK-mediated cytotoxicity (Figure 1).

### 3.2. Soluble Factors

In addition to ligands expressed on the leukemia cell surface, several NK activating or inhibiting ligands can be secreted or shed from the cell surface. These ligands inhibit NK functions with the blockade or destruction of their corresponding receptors. In CLL, several studies have identified soluble NKG2D ligands in patient serum, with MICA, MICB, and ULBP2 being the most commonly upregulated [110,111,121,141]. In one report, 100% of CLL patients had at least one ligand elevated in serum, and over 90% had multiple present [110]. These increase with later disease stages or correlate with shorter treatment-free survival [110,111,121]. These soluble ligands decrease NK reactivity by triggering NKG2D endocytosis and destruction [110,142]. Similarly, CLL patient serum contains elevated levels of BAT3 (NKp30 ligand), HLA-E (NKG2A ligand), HLA-G (ILT2/KIR2DL4 ligand), and Gal-9 (TIM3 ligand)—these soluble ligands have all been demonstrated to have NK-inhibitory effects [97,103,121,128,135,143,144,145,146].

CLL cells produce multiple cytokines that modulate the immune environment, including NK cells. For instance, CLL cells have long been known to produce TGFβ (transforming growth factor beta) [147,148]. While its specific role in CLL-induced NK inhibition has not been established, TGFβ inhibits NK cells by decreasing the expression of activating receptors and cytotoxic molecules, altering NK metabolism, and inhibiting cytokine production [149,150,151]. Similarly, CLL cells produce IL-10 and adenosine, which inhibit NK cells but have not been directly studied in CLL-induced NK dysfunction [152,153,154,155]. In contrast, CLL patients also have increased serum levels of IL-2, IL-12, and IFNα, which stimulate NK cells and may counteract the suppressive effects of other cytokines [21]. Beyond these examples, CLL patients have altered levels of various other cytokines and chemokines, which may exert additional effects on patients’ NK cells [21].

### 3.3. Altered Immune Populations

Finally, CLL cells induce alterations in other cell populations that may impair NK function. CLL patients have increased populations of regulatory T cells (Tregs) and myeloid-derived suppressor cells (MDSCs), both of which suppress NK cells [26,27,85,90,153,156,157,158,159,160,161,162,163,164,165,166,167]. Nurse-like cells (NLCs) are monocyte-derived cells that are recruited and differentiated by CLL cells; they, in turn, help support the leukemic cells’ survival and proliferation [42]. While these cells’ effects on NK cells have not been studied directly, nurse-like cells closely resemble M2-polarized macrophages, which do have known NK-suppressive effects [42,168].

## 4. CLL Resistance to NK Cytotoxicity

In addition to inducing natural killer cell dysfunction, multiple studies have shown that CLL cells are resistant to direct cytotoxicity caused by autologous or allogeneic NK cells [95,97,100,105,106,169,170]. CLL cells’ inability to compete with K562 or Daudi cells in cold target inhibition assays and a lack of NK-CLL conjugate formation in vitro suggest that this resistance is due to a lack of target recognition by NK cells [100,105]. However, the addition of targeting antibodies such as rituximab, obinutuzumab, ofatumumab, or alemtuzumab can induce recognition and cytotoxicity [97]. After target recognition, effective cytotoxicity by NK cells requires the formation of an immunological synapse. Therefore, the reduced expression on CLL cells of LFA-1 and ICAM-1 (two key adhesion proteins for binding to NK cells) seen in one study may further increase the CLL evasion of natural killer cells [100]. However, data on this is mixed, with another study finding a nonsignificant decrease in ICAM-1 but a nonsignificant increase in LFA-3 on CLL cells [97]. Finally, it was recently demonstrated that CLL cells reorganize their cytoskeletons after immunological synapse formation [171]. This reorganization causes an accumulation of actin filaments at the immunological synapse, ultimately leading to the decreased uptake of granzyme B and resistance to NK cytotoxicity [171].

Counterintuitively, some signals from NK cells may support CLL cell survival and NK resistance. For instance, BAFF (B cell activating factor) secretion by activated NK cells can decrease CLL cell susceptibility to NK cells [172]. NK-derived interferon-gamma (IFNγ) can have varied effects, including inhibiting CLL cell apoptosis and upregulating HLA-E [117,173]. Interactions between 4-1BB on NK cells and 4-1BB ligand (4-1BBL) on CLL cells have been demonstrated to have a surprising inhibitory effect on NK reactivity through the “reverse signaling” of 4-1BBL [174].

## 5. Effects of CLL Therapeutics on NK Function

### 5.1. BTK Inhibitors

Since its approval for marketing, ibrutinib has rapidly become a treatment of choice for CLL. Intended as a targeted inhibitor of Bruton’s tyrosine kinase (BTK) in the B cell receptor signaling pathway, ibrutinib also binds and inhibits several other related kinases, including IL-2-inducible T cell kinase (ITK) [175]. Ibrutinib induces CLL cell apoptosis, inhibits survival signals from external stimuli, inhibits tissue homing, and decreases CLL chemokine production [176,177,178]. In addition to its antileukemia effects, ibrutinib alters the host immune system, as exemplified by patients’ decreased infection risk (although with some new risk of opportunistic infections) and ibrutinib’s efficacy and FDA approval for chronic graft-versus-host disease [179,180,181].

Because both BTK and ITK are involved in NK cell activation, it is unsurprising that ibrutinib has significant effects on NK function (Figure 3) [182,183,184]. In fact, several studies have demonstrated decreased ADCC activity caused by NK cells treated with ibrutinib [183,185,186,187,188,189]. This may be exacerbated by ibrutinib’s ability to decrease CLL CD20 expression [185,190,191], although the evidence linking CD20 density to ADCC activity is inconsistent [102,192]. Interestingly, ibrutinib does not seem to have a significant effect on antibody-independent direct cytotoxicity, likely because of the different roles of ITK in Fc receptor signaling versus other NK receptors [182,189]. In addition to its effects on cytotoxicity, ibrutinib inhibits NK cell activation-induced death and slightly decreases proliferation after IL-2 + IL-15 stimulation in vitro [193,194]. In addition to its effects on NK activation, long-term ibrutinib treatment has been shown to decrease levels of CD16-negative NK cells without altering CD16-positive NK counts [160]. Ibrutinib also has varied effects on the immune system that are likely (though not proven in CLL) to alter NK function. These include decreasing CLL IL-10 production [193,195] and PD-L1 expression [195], decreasing Tregs [160,193,196] and MDSCs [160], altering T cell function and populations [193,197,198,199], altering monocyte/macrophage activity [200], and altering systemic cytokine levels [198]. In agreement with ibrutinib’s ability to inhibit rituximab-dependent effector functions by NK cells and other cells [183,185,186,187,188,189,200,201,202], the addition of rituximab to ibrutinib does not improve response rates or progression-free survival in CLL, although it does quicken responses and lower residual disease levels [203,204].

Acalabrutinib is a second-generation BTK inhibitor with increased specificity for BTK without inhibiting ITK [205]. Accordingly, acalabrutinib does not significantly decrease ADCC or direct cytotoxicity by NK cells [188,189,205]. However, it also lacks the ability to protect NKs from activation-induced cell death [193]. In concordance with preserved antibody effector functions, the addition of obinutuzumab to acalabrutinib has recently been associated with better progression-free survival, the overall response rate, the rate of complete response, and a trend toward improved overall survival in a post hoc analysis [206].

### 5.2. PI3K Inhibitors

Idelalisib is an inhibitor of PI3Kδ (phosphoinositide 3-kinase delta), another kinase in the B cell receptor signaling pathway. In addition to its role in B cells, PI3K also mediates NK activation through various receptors [207]. Similar to BTK inhibitors, idelalisib also inhibits NK cytotoxic function, although this effect is weaker than seen with ibrutinib and has not been consistently found in all studies (Figure 3) [185,186,187,189,208,209,210]. Idelalisib also inhibits NK proliferation after IL-2 or IL-2 + IL-15 stimulation in vitro [194,210]. Idelalisib has diverse immunomodulatory effects and is associated with significant immunologic adverse events, likely related to the depletion and inhibition of regulatory T cells [211,212,213,214]. While this may suggest indirect effects on NK cell function, the in vivo effects of idelalisib on NK function have not been studied. Similar to ibrutinib, idelalisib treatment in vitro decreases CLL CD20 expression [185]. Duvelisib, another PI3K inhibitor, has also been found to mildly inhibit ADCC [189].

### 5.3. Venetoclax

Venetoclax, an inhibitor of Bcl-2 (B cell lymphoma 2, an anti-apoptotic protein in the intrinsic pathway of apoptosis), is not thought to directly interact with immunoreceptor signaling, but it does exert a significant immunologic effect through the depletion of leukocytes, including neutrophils, T cells, and NK cells (Figure 3) [85,215]. Venetoclax is cytotoxic to NK cells in vitro [216,217], and treatment with venetoclax + obinutuzumab or venetoclax + ibrutinib depletes NK cells in patients [85]. This depletion predominantly affects resting NK cells, because NK stimulation leads to the upregulation of Bcl-2, Bcl-XL, and Mcl1 (anti-apoptotic proteins in the Bcl-2 family) with the downregulation of Bim and Bid (pro-apoptotic proteins in the Bcl-2 family) [216,218,219,220,221,222,223]. Accordingly, the remaining NK cells after venetoclax + obinutuzumab treatment show increased activation [85]. Finally, like other therapies, venetoclax is likely to have indirect effects on NK function, for instance, by decreasing Treg populations [85].

### 5.4. Antibodies

While the role of NK cells in response to anti-CLL antibody therapy has been well-known for years, the effects of these therapies on NK cells is a more recent development (Figure 3) [69,70]. Rituximab was the first anti-CD20 antibody used for CLL and has long been a mainstay of treatment for this disease. In vitro, exposure to rituximab-coated targets leads to CD16 downregulation and the subsequent decreased ADCC function of NKs [224]. Surprisingly, it also leads to a decrease in cytotoxic function mediated by other NK receptors such as NKG2D, DNAM-1, 2B4, and NKp46 [224]. Ofatumumab, another anti-CD20 antibody with increased complement activity, has shown a similar ability to decrease NK CD16 expression and cytotoxic activity in vitro [224,225]. Additionally, studies have found that the stimulation of monocytes or neutrophils with rituximab or ofatumumab could induce reactive oxygen species production, thus inhibiting the cytotoxic activity of cocultured NK cells and inducing their apoptosis [226,227]. This effect can be prevented by treatment with idelalisib to prevent monocyte ROS production [209]. Obinutuzumab is an anti-CD20 antibody that has been glycoengineered for nonfucosylated glycoforms at Asn297, leading to increased binding to FcγRIIIa and ADCC in vitro [189,192,228]. In addition to greater cytotoxicity, obinutuzumab is able to overcome inhibitory signals from KIR-HLA interactions better than rituximab [229]. In vitro, obinutuzumab’s stimulation of NK cells leads to a subsequent increased responsiveness to cytokine stimulation but decreased cytotoxic activity and IFNγ release mediated by various NK receptors [230,231]. Stimulation with obinutuzumab- or rituximab-coated target cells can induce the expansion of memory-like NK cells [232]. In patients, obinutuzumab treatment is associated with decreased peripheral NK counts and an increase in the ratio of CD56^bright^ vs. CD56^dim^ NK cells [85,233,234,235]. This decline happens rapidly during the course of the first obinutuzumab infusion, and NK counts remain persistently low between treatments [233]. While the rapidity of this response may suggest NK migration rather than depletion, this has not been demonstrated experimentally. A study of rituximab in follicular lymphoma demonstrated a decrease in peripheral NK counts while inducing NK proliferation within blood and lymph nodes, but it did not find an increase in lymph node NK cells [236]. Similar to in vitro results, this study found a temporary decrease in NK ex vivo responsiveness to rituximab-coated targets [236]. Initial infusion with obinutuzumab is associated with a significant spike in several pro-inflammatory cytokines that may influence the resulting NK cell reaction [233]. Finally, the anti-CD52 antibody alemtuzumab has also demonstrated efficacy in CLL, though it was associated with lower efficacy and greater adverse effects when compared with rituximab in a phase III trial [237]. Alemtuzumab administration depletes NK cells, which express CD52 [87].

### 5.5. Cytotoxic Chemotherapy

CLL has long been treated with cytotoxic chemoimmunotherapy regimens such as FCR (fludarabine, cyclophosphamide, and rituximab) therapy or BR (bendamustine and rituximab) therapy. These treatments are very lymphodepleting, and, in fact, cyclophosphamide and fludarabine are often intentionally used for this purpose before HSCT, CAR-T (chimeric antigen receptor T cells), or other adoptive cell therapies. Accordingly, NK counts decrease during both FCR and BR therapies, although NKs are generally less susceptible and recover faster than T cells (Figure 3) [238,239,240]. These regimens have rapidly fallen out of favor in the US after multiple phase 3 trials demonstrated improved progression-free survival by using targeted therapeutics such as ibrutinib, acalabrutinib, and venetoclax combined with anti-CD20 antibodies [203,206,241].

### 5.6. Lenalidomide

Although not a routine part of CLL clinical management, the immunomodulatory drug lenalidomide (a derivative of thalidomide that binds to cereblon and induces ubiquitination and the degradation of multiple targets such as IKZF1/Ikaros and IKZF3/Aiolos) has demonstrated activity in CLL, as well as significant stimulatory effects on natural killer cells (Figure 3) [242]. Lenalidomide increases NK activation, proliferation, cytokine production, and cytotoxicity both by acting on NKs themselves and by stimulating T cells or dendritic cells [93,113,150,243,244,245,246,247]. Furthermore, lenalidomide prevents NK inhibition caused by IL-6 and TGFβ [150]. Notably, this NK stimulation increases the direct cytotoxicity and ADCC of primary CLL cells [93,245]. Additionally, lenalidomide treatment alters CLL cells by increasing CD20 expression, decreasing MHC-I, increasing HLA-E, and decreasing the rituximab concentration needed to activate NKs [113,246,247].

## 6. Advantages of NK Therapy for CLL

So far, most preclinical and clinical effort on adoptive cell therapies for CLL has focused on T cell-based therapies given the successes of CAR-T therapy for acute lymphoblastic leukemia, diffuse large B-cell lymphoma, and mantle cell lymphoma. However, CAR-T therapy has not been as successful in CLL, with only a fraction of patients achieving a durable response [248]. Natural killer cell-based therapies offer several advantages over T cell therapies such as CAR-T, which may lead to greater success in treating CLL. Unlike T cells, NK cells recognize and kill targets through an MHC-independent mechanism. Therefore, using HLA-matched samples (for CAR-T, generally autologous) is not necessary to avoid graft-versus-host reactions associated with HLA-mismatched T cells. This enables the use of “off-the-shelf” NK cells, which are expanded from an established donor pool and stored for use as needed rather than expanded on a case-by-case basis. These cells can additionally be optimized for the best KIR profile or KIR mismatch between the donor cells and recipient. Natural killer cells also have the advantage of recognizing tumor cells through both antigen-dependent (via combined treatment with anti-CLL antibodies) and antigen-independent mechanisms (the innate NK receptors). This decreases the risk of resistance caused by antigen loss, as seen with CD19-CAR-T therapy [249]. NK cells can be genetically modified to insert a CAR, but the availability of antibody-mediated antigen recognition may increase flexibility and decrease costs and time to treatment. Finally, the safety profile of NK cell therapies has been very favorable, with various formulations of NK cells being administered in multiple trials without major safety concerns [250]. Notably, T cell therapies tend to produce high levels of IL-6, leading to potentially fatal complications of cytokine release syndrome and neurotoxicity—because NK cells produce much less IL-6 than T cells, this risk is greatly diminished [251,252]. Because NK cells (and CLL-targeting antibodies) have a finite lifespan, there is also the potential to avoid the prolonged B cell lymphopenia and hypogammaglobulinemia seen in many patients after CD19-CAR-T therapy, although this theoretical benefit will need to be tested in clinical trials [253].

## 7. Previous and Ongoing Attempts at NK Therapy for CLL

NK therapies for CLL have a long history of development with only recent efforts showing promise. The first wave began in the late 1980s with the stimulation of peripheral blood mononuclear cells (PBMCs) using IL-2, IFNα, and/or anti-CD3 to produce lymphokine-activated killer cells (LAKs) [100,108,254,255,256,257]. While two studies found that the normal cytotoxic function of patient-derived cells could be restored [108,256], most demonstrated a persistent cytotoxic defect associated with CLL even after stimulation [100,254,255]. Furthermore, primary CLL leukemia cells are resistant to both patient-derived and healthy-donor-derived LAKs [100,254,255]. More recently, various cytokine protocols have been shown to effectively stimulate CLL patient NK cells in vitro, including IL-2, IL-15, IL-21, IL-12 + IL-15, and IL-2 + IL-18 [91,93,96,97,98,258]. These studies have shown the potential to revert patient NKs to normal or increased activity, although none have been used to expand the populations for use as an actual therapy, and some studies do show a persistent defect in NK activity [98]. The stimulation of allogeneic NK cells with an Epstein–Barr virus (EBV)-positive lymphoblastoid cell line (EBV-LCL) has been shown to induce anti-CLL cytotoxicity in vitro, particularly against poor-prognosis (p53 mutated, IGHV unmutated) patient samples [170,259,260,261]. CD19 or CD20 chimeric antigen receptor NK cells (CAR-NK) also show significant potential for CLL. This strategy was first demonstrated using the NK-92 cell line, which showed the CAR-dependent killing of CLL in vitro [262,263]. This was improved by using HLA-mismatched NKs expanded from cord blood and transduced with anti-CD19 CAR, IL-15, and an inducible caspase 9. These CAR-NKs show effective expansion, cytotoxic activity, and NK survival in vitro and in vivo [264]. Notably, however, these authors found that patient-derived CAR-NKs had inferior cytotoxic activity in vitro in comparison with cord blood-derived CAR-NKs [264]. This work was translated to a small phase 1–2 trial of 11 patients with non-Hodgkin lymphoma or CLL, which demonstrated the safety of the NK cell product without graft-versus-host disease (GVHD), cytokine release syndrome, or neurotoxicity (major concerns with CAR-T therapy) [252]. The main adverse events were cytopenias, which were likely related to the fludarabine/cyclophosphamide conditioning regimen, although the contribution of the NK cells could not be excluded [252]. Most (4/5) CLL patients in this study achieved objective responses, although it is difficult to distinguish the effects of the NK cell treatment from the conditioning regimen, and most patients quickly moved to subsequent treatment after receiving NKs [252]. Similarly, another group demonstrated the in vitro anti-CLL cytotoxicity of a CAR-NK expressing CD19 CAR, high-affinity non-cleavable CD16, and a membrane-bound IL-15/IL-15R fusion molecule [265]. Most recently, our lab tested the expansion of NK cells using membrane-bound IL-21 (mbIL-21) expressing feeder cells, demonstrating that this stimulation produces high numbers of NK cells with potent anti-CLL activity in vitro and in vivo, including both allogeneic and autologous targets [266]. The administration of a targeted antibody here is required.

## 8. Conclusions

While overshadowed by the focus on T cells, natural killer cells play a vital role in anticancer immune defenses. Extensive evidence from CLL and other hematologic malignancies indicates that NK cells can play an important role in treating leukemia in the context of HSCT, antibody therapy, or NK therapy itself. While the body of literature cited in this review does reflect a broad array of inhibitory effects directed from CLL cells toward NK cells and a complex network of interactions, successful treatments and NK stimulations in the clinic and laboratory indicate that these inhibitory effects can be overcome. With recent developments such as CD19-CAR-NK and mbIL-21-expanded NK therapy, the future of NK therapy for CLL looks bright.

## Figures and Tables

**Figure 1 cancers-14-05787-f001:**
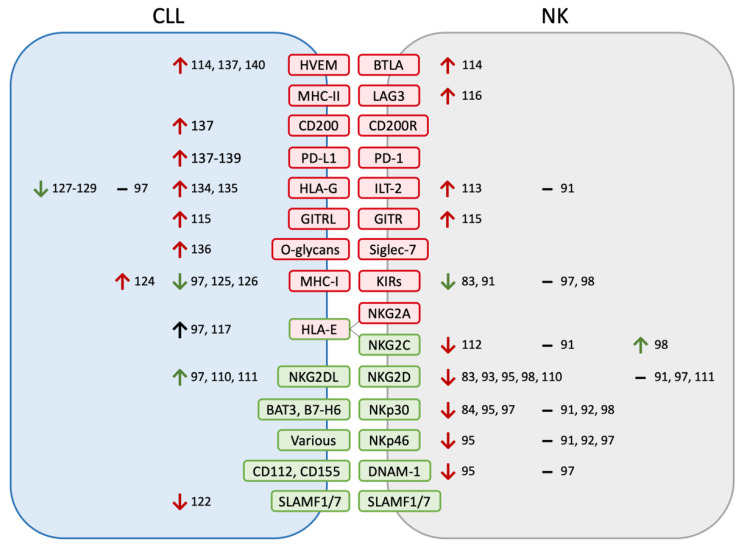
Altered CLL-NK ligand–receptor interactions. Ligand–receptor pairs are color-coded by their activating (green) versus inhibitory (red) effects on NK cells. Changes are listed as increases (↑), decreases (↓), or no change (–) and are similarly color-coded according to whether the measured change is expected to increase or decrease NK activation, as well as the relevant citations listed. Different studies have found varying (and, at times, conflicting) changes in ligand and receptor expression levels that broadly point to decreased NK activity against CLL. NKG2DL refers to the ligands of NKG2D (MICA, MICB, ULBP1-6).

**Figure 2 cancers-14-05787-f002:**
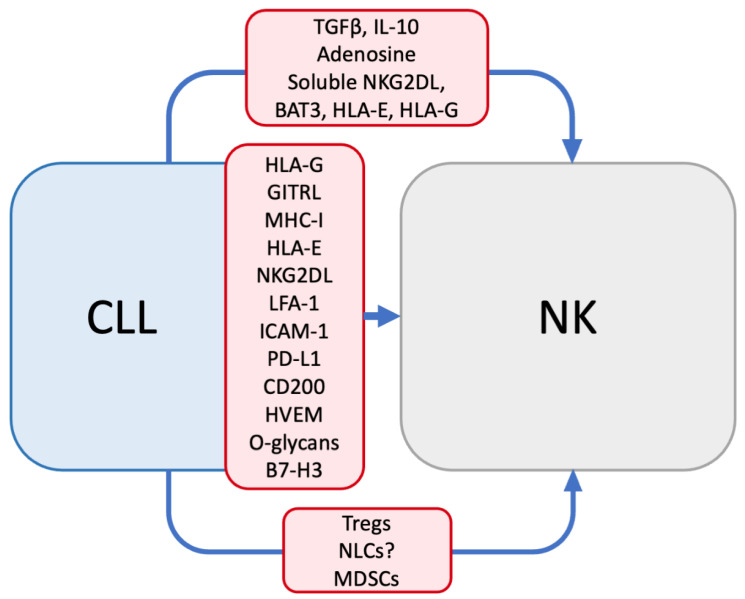
Mechanisms of CLL-induced NK suppression. NK dysfunction seen in CLL patients is ascribed to multiple mechanisms, including soluble factors (cytokines, metabolites, ligands), surface ligand expression, and altered immune populations (Tregs, NLCs, MDSCs).

**Figure 3 cancers-14-05787-f003:**
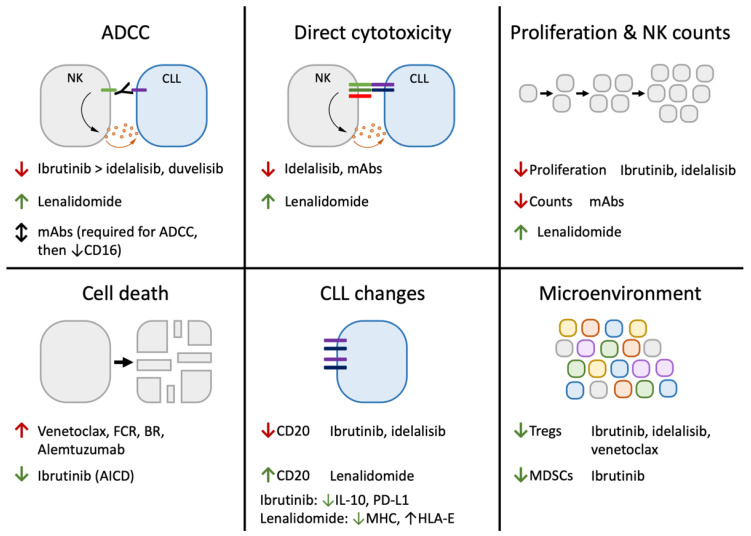
Key effects of CLL therapeutics on NK function. Current CLL therapeutics have been demonstrated to affect natural killer cells in a variety of ways. This includes directly increasing or decreasing NK cytotoxic function, proliferation, and death. It also includes indirect methods via these therapeutics’ effects on CLL and other cells in the microenvironment. Changes are listed as increases (↑) or decreases (↓), and are color-coded according to whether the measured change is expected to increase (green) or decrease (red) anti-leukemia NK activity.

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
