# Peer review of "Natural Killer Cells in Chronic Lymphocytic Leukemia: Functional Impairment and Therapeutic Potential"

_cancers, 2022, doi:10.3390/cancers14235787_

Round 1

Reviewer 1 Report

See attached pdf. 

Reviewer 2 Report

The authors present an interesting article about NK cells in CLL. They propose a narrative with a strong focus on possible practical aspects related to NK cells for the treatment of haematological neoplasms in general rather than CLL in particular.

The article requires significant editorial changes - in its current form it is chaotic and suffers from relatively low-quality of language. It is also highly speculative in numerous places - in some parts this is acceptable, but in others, the narration is so complicated that one cannot distinguish which parts are based on direct experimental data from CLL and which are based on other diseases or studies not directly related to NK cells.

It would be helpful if authors could briefly summarise the effects of various drugs on NK cells in the form of either a table or a figure.

This manuscript may be an important contribution to the field if authors significantly improve quality of language and organise text to avoid chaos.

Reviewer 3 Report

In this review, Yano et al. has described characteristics and functions of NK cells in chronic lymphocytic leukemia. Some issues should be addressed.

1. In the title, specify NK and CLL. Abbreviations should be avoided in the title.

2. Some parts are redundant, such as 1.2 and 1.3.

3. For some paragraphs, there are not conclusive statements. For example, in paragraph 2.2, it is not clear what is the normal NK phenotype.

4. Figura 1 is very informative; however, little of this information is found in the manuscript.

5. Paragraph 6 is a little bit out of contenxt in the view of CLL. Please consider to shorten it.
